# Be Selfish, But Wisely: Investigating the Impact of Agent Personality in Mixed-Motive Human-Agent Interactions

**Kushal Chawla**[1]   **Ian Wu**[1]   **Yu Rong**[1]   **Gale M. Lucas**[2]   **Jonathan Gratch**[2]

University of Southern California, Los Angeles, USA

[1]{kchawla,ianwu,yrong016}@usc.edu
[2]{lucas,gratch}@ict.usc.edu

## Abstract

A natural way to design a negotiation dialogue system is via self-play RL: train an agent that learns to maximize its performance by interacting with a simulated user that has been designed to imitate human-human dialogue data. Although this procedure has been adopted in prior work, we find that it results in a fundamentally flawed system that fails to learn the value of compromise in a negotiation, which can often lead to no agreements (i.e., *the partner walking away without a deal*), ultimately hurting the model's overall performance. We investigate this observation in the context of DealOrNoDeal task, a *multi-issue negotiation* over *books*, *hats*, and *balls*. Grounded in negotiation theory from Economics, we modify the training procedure in two novel ways to design agents with diverse personalities and analyze their performance with human partners. We find that although both techniques show promise, a selfish agent, which maximizes its own performance while also avoiding walkaways, performs superior to other variants by implicitly learning to generate value for both itself and the negotiation partner. We discuss the implications of our findings for what it means to be a successful negotiation dialogue system and how these systems should be designed in the future.

## 1 Introduction

*"Firms [Agents], in the pursuit of profits, are led, as if by an invisible hand, to do what is best for the world." - Adam Smith: The Father of Modern Economics*

Negotiation is a crucial social influence interaction (Chawla et al., 2023), ubiquitous in everyday scenarios, from deciding who performs household chores to high-stakes business deals and legal proceedings. Consequently, negotiation dialogue systems find numerous applications in advancing conversational AI assistants (Leviathan and Matias,

| Context (Alice: RL-Based, Bob: Supervised) | |
|---|---|
| Counts | Book = 2, Hat = 1, Ball = 3 |
| Alice Values | Book = 1, Hat = 2, Ball = 2 |
| Bob Values | Book = 0, Hat = 7, Ball = 1 |
| **Dialogue** | |
| Alice | i would like the balls and hat and a book |
| Bob | you can have the balls and one book |
| Alice | i will take the balls and hat |
| Bob | deal |
| Alice | <dealselection> |
| **Output** | |
| Alice | Book = 0, Hat = 1, Ball = 3 |
| Bob | Book = 2, Hat = 0, Ball = 0 |
| **Reward** | |
| Alice | 8/10 |
| Bob | 0/10 |

Table 1: A sample problematic negotiation dialogue between the standard RL agent (Alice) and a supervised model (Bob), based on Lewis et al. (2017). The task here is to divide the available books, hats, and balls between the two players. In this case, Bob accepts a deal even though it is very unfavorable, resulting in a high score for Alice.

2018), by advising human decision-making (Zhou et al., 2019), and in pedagogy, by making social skills training more effective (Johnson et al., 2019).

Negotiation is a complex *mixed-motive interaction*, involving motivations for both self-serving as well as cooperative and socialistic behaviors. A successful negotiator must not only learn to *extract concessions* from the partner but also to *make concessions* in order to reach an agreement. Maintaining this balance between self-interest and the interests of negotiation partners makes it a challenging task for automated dialogue agents. If an agent tries to take too much without any compromise, this can push the partner to walk away without an agreement, hurting the outcomes for both players.

One natural way to design such a system is through Self-play Reinforcement Learning (RL). **Step I:** Train a model $S$ that imitates human-human dialogue data in a supervised manner. **Step II:** Create two copies of $S$, $S_{RL}$, which is the initialization

for the RL agent, and $S_{US}$, which acts as a *fixed simulated user*. **Step III:** Update $S_{RL}$ to maximize its performance using an online RL algorithm by making it interact with $S_{US}$ (*bot-bot interactions*) and recording the final performance achieved by the model (the *reward*).

Although adopted in prior work (Lewis et al., 2017; He et al., 2018), we argue that this procedure leads to a fundamentally flawed system that fails to learn the value of compromise in a negotiation. **Arguments: 1)** The available human-human negotiation data mainly contains dialogues that end in agreements ($\approx 80\%$ in DealOrNoDeal dataset (Lewis et al., 2017)), instead of walkaways or no agreements, leading to a highly prosocial simulated user $S_{US}$ that tends to show agreement, regardless of how favorable the deal is. Hence, when training the RL agent $S_{RL}$ to maximize its own performance against $S_{US}$, $S_{RL}$ becomes highly self-interested without learning to make any concessions since that leads to a high reward for $S_{RL}$. We show one such problematic conversation between these two models in Table 1. **2)** Another piece of evidence comes from prior work (Lewis et al., 2017). Even though such an RL model seems to perform well in automated evaluations (against the simulated user), it performs much worse against human partners, who often prefer to walk away with no agreement and 0 points earned for both parties rather than agreeing to an uncompromising partner. **3)** Finally, one can look at what happens if $S_{RL}$ is made to play with another copy of $S_{RL}$. In this case, we find that the agents simply get stuck - both continuously asking what they want without looking for a compromise (refer to Appendix A for a sample conversation).

This failure hurts the practical utility of the system, both from the perspective of being a successful negotiator in conversational AI use cases and for providing social skills training in pedagogy. The key challenge here is to somehow teach the model to be a *mixed-motive* negotiator instead of only self-interested, with a better understanding of the concept of walkaways in a negotiation, even though the collected dialogue data primarily consists of dialogues ending in agreements. To address this, we investigate two modifications to the training procedure, resulting in systems that exhibit diverse personalities[1]: 1) We vary the RL reward directly so

that the model is forced to take the partner's interests into account. This corresponds to manipulating the *motives* of the dialogue agent, a psychological concept that has received significant attention in the literature (Murphy and Ackermann, 2014). For this purpose, we rely on a *a measure of utility* from negotiation theory in Economics (Fehr and Schmidt, 1999), which helps us to control selfish vs. fair behavior explicitly. 2) We vary the *personality of the simulated user* that the RL agent is trained with. This approach essentially manipulates the interaction experience that the agent receives so that the agent is itself allowed to discover the value of making concessions by being better exposed to walkaways during training. We now summarize our contributions:

1. We provide evidence that the standard self-play RL training procedure fails to develop sophisticated negotiation dialogue systems useful in practical scenarios (Section 1).

2. To address this issue, we devise novel ways to modify the training procedure, grounded in negotiation theory from Economics, so as to design systems that exhibit diverse personalities and better understand the concept of walkaways (Section 3).

3. Through a comprehensive automated and human evaluation, we investigate what model variation allows for superior performance. Our key finding is that a selfish agent, which maximizes its own performance while also avoiding walkaways, achieves superior performance to other variants by learning to generate value for both itself and the negotiation partner (Section 5).

4. We discuss the implications of our findings for designing and evaluating negotiation dialogue systems in the future (Section 6).

## 2 Related Work

Historically, negotiation has been studied across several disciplines, including Game Theory (Nash, 1950) and Psychology (Adair et al., 2001). More recently, there has been an increasing interest in human-agent negotiations as well (Baarslag et al., 2016; Gratch et al., 2015). Extensive research has examined the effects of both agent and human personality in negotiation and related decision-making

---

[1]By personality, we simply refer to the consistent behavior portrayed by the trained agent (https://www.apa.org/topics/personality)

tasks (Bogaert et al., 2008; Mell et al., 2018; van Wissen et al., 2009). However, most prior efforts analyze interactions based on structured communication channels such as through a menu of options (Mell and Gratch, 2016). Instead, Beaunay et al. (2022) studied participants' extreme reactions to unfair offers by a selfish chatbot in an ultimatum game. We contribute to this line of research by exploring diverse dialogue agent personalities and studying their impact on negotiation performance.

Several dialogue datasets (Lewis et al., 2017; Chawla et al., 2021; He et al., 2018; Yamaguchi et al., 2021) have fueled research into designing negotiation dialogue systems. RL has been a popular technique of choice in this space (Zhang et al., 2020; Yang et al., 2021). Yang et al. (2021) modeled the personality of the partners by a one-step dialogue-act look ahead in a buyer-seller negotiation domain and found that it leads to a higher agreement rate. Complementary to this, our work investigates the impact of diverse agent personalities by modifying both the underlying reward and the partner personality for RL training. In addition, we focus on using selfplay RL directly at the utterance level, which does not need additional annotations or separate parser and generator modules that are relatively difficult to design for general multi-issue negotiation tasks.

Other recent work has also explored the incorporation of additional annotations such as dialogue acts and strategy labels (Joshi et al., 2020). Nevertheless, our paper focuses on designing agents for mixed-motive interactions, which is fundamental to any underlying negotiation context and model architecture.

## 3 Methodology

We focus on bilateral multi-issue negotiations which involve a fixed set of issues (e.g., *books*, *balls*, and *hats* in the DealOrNoDeal dataset (Lewis et al., 2017)). Each issue has a predefined quantity along with a random value (potentially different) assigned for every player. The players engage in a dialogue to reach an agreement – a possible division of all the available items in which they try to maximize the total value of the items that they get.

Our goal here is to develop techniques so that the trained dialogue models learn to make concessions (e.g., by offering deals that help the partner) for their partners apart from just learning to extract concessions from them. As discussed earlier, this

mixed-motive behavior is a fundamental expectation from a practical negotiation dialogue system. To achieve this, we propose two complementary techniques – first, where we *explicitly* incorporate the partner's performance into the reward function of the RL agent, and second, where the model *implicitly* learns to make concessions by interacting with a specific partner during training. We start by describing our base RL framework and then discuss the two proposed techniques.

### 3.1 Self-play RL for Negotiation Dialogue

We use the Selfplay RL framework introduced by Lewis et al. (2017) for training negotiation dialogue systems. Their pipeline consists of first training a supervised agent to mimic the collected human-human dialogue data and then using selfplay RL to further optimize the model. As Lewis et al. (2017) note, training a supervised agent to mimic human actions is a scalable and domain-agnostic starting point. However, this model by itself is unable to engage in strategic actions necessary for effective negotiation. By then having the supervised model negotiate with a fixed copy of itself (simulated user) and fine-tuning the model using an online RL algorithm, the model can be optimized towards a given reward function (in this case, the points scored by the agent in the negotiation).

The framework relies on a sequence-to-sequence model based on an ensemble of Gated Recurrent Units or GRUs (Cho et al., 2014). The model consists of one unidirectional GRU for encoding the input goals of the agent, another to encode the utterances from both the agent and the human partner, and one bidirectional GRU to generate the output deal once the negotiation is over.[2]

In the supervised stage, the model is trained on a combined cross-entropy loss that jointly optimizes both the next-token prediction and the output deal prediction. The RL agent is trained with the REINFORCE method (Williams, 1992).

### 3.2 Proposed techniques

#### 3.2.1 Varying the reward function

The key idea here is to incorporate the partner's performance into the reward function used for training the RL agent. Intuitively, this would make the agent

---

[2]Although the exact choice of the model architecture is irrelevant to our analysis, we choose this lightweight architecture to enable our analysis with different kinds of agent personalities.

more prone to offering deals or accepting deals that help the partner as well.

To approach this systematically, we leverage a *measure of utility* defined in negotiation theory in Economics by Fehr and Schmidt (1999). The utility function $U_i(x)$ is defined as follows:

$$U_i(x) = x_i - a * (max(0, x_j - x_i)) \\ - b * (max(0, x_i - x_j)) \quad (1)$$

where $b \leq a, 0 \leq b < 1$. $i$ and $j$ denote the two players in the negotiation. $x = (x_i, x_j)$ denotes the points scored by the corresponding players. $U_i(x)$ essentially captures the utility gained by the player $i$ from the negotiation, given the points scored by all the players $(x)$.

Fehr and Schmidt (1999) defined this utility measure to model diverse behaviors in human-human negotiations, noting that merely assuming that all players are selfish does not explain the data. Hence, to capture the diversity in human behaviors, the equation includes additional terms that capture the *advantage* and the *disadvantage* of player $i$ with respect to player $j$ in the negotiation. We repurpose this utility measure directly as the reward for the RL agent. By varying the coefficients $a$ and $b$, different reward functions that promote diverse personality behaviors can be generated. We demonstrate this in Table 2. For our analysis in this paper, we choose the *selfish* and *fair* configurations.

### 3.2.2 Varying the negotiation partner

While the above method, in some ways, explicitly pushes the agent to take the partner's performance into account, we now propose another technique to achieve this more implicitly.

Since the supervised model tends to show socialistic behaviors (Table 1), the RL agent fails to explore scenarios that do not lead to an agreement and, hence, cannot capture the notion of walkaways in the learned policy. However, if the agent were to interact with an uncompromising partner, this could be leveraged to simulate "walkaways" during model training, with the hope that the model discovers ways to avoid disagreements (while still optimizing on the reward), and thus implicitly learns about making concessions for the partner.

Hence, the key idea here is to vary the personality of the partner model. In addition, we define a length cut-off $l$: if the conversation reaches $l$ utterances, this is seen as a disagreement, and both agents receive 0 points from the negotiation. We

explain how we design the diverse partner personalities for training later in Section 4.

## 4 Experimental Design

We proposed two ways of training dialogue models that capture the mixed-motive nature of negotiations: 1) explicitly, by varying the reward function for the RL algorithm (Section 3.2.1), and 2) implicitly, by varying the partner with which the RL model is trained (Section 3.2.2). **The primary research question we aim to answer is what variation leads to superior performance with human partners.** We first describe the dataset and the study design, followed by results in Section 5.

**Dataset**: We use the DealOrNoDeal dataset (Lewis et al., 2017), which is based on the Multi-Issue Bargaining Task (Fershtman, 1990) design. The dataset uses a simplistic design involving 3 issues (*books*, *hats*, and *balls*), and has been a popular choice for research in negotiation dialogue systems. It comprises 5808 dialogues in English based on 2236 unique scenarios, where a scenario refers to the available items up for grabs and their corresponding values for the two players. In each scenario, there is a fixed quantity of each issue, and players are randomly assigned a point value before the negotiation for each of the 3 issues. The goal of the dialogue is to reach an agreement on the possible division of all the available items, where each player strives to maximize the total value of the items that they get. The maximum possible value for a player is 10. However, if no agreement is reached, then both players end up with 0 points. Nearly 80% of the dialogues end in agreement, with an average of 6.6 turns per dialogue and 7.6 words per turn. We use the same splits as the original dataset paper to train our dialogue agents.

**Study Design**: We design a 2 X 3 study based on the strategies described in Section 3. We use a three-stage process to develop the 6 agent personalities: **Stage 1**: Develop a supervised likelihood model, following Lewis et al. (2017). **Stage 2**: Train two RL dialogue agents by varying the reward using the *selfish* and *fair* utility functions selected from Table 2. Note that the selfish configuration here is equivalent to the base RL model trained by Lewis et al. (2017). **Stage 3**: Train the remaining four RL agents by varying the reward function (*selfish* vs. *fair*) and using either of the two models trained in Stage 2 as partners. We provide an overview of this process and describe our

| a | b | Utility ($U_i(x)$) | Interpretation |
|---|---|---|---|
| 0 | 0 | $x_i$ | Selfish: partner points don't matter. |
| 1 | 0 | $x_i - (max(0, x_j - x_i)$ | Doesn't like if the partner outperforms. |
| 0 | -1 | $x_i + (max(0, x_i - x_j)$ | Selfish and Envious (desires poor partner performance) |
| 0.75 | 0.75 | $x_i - 0.75 * max(0, x_j - x_i) - 0.75 * (max(0, x_i - x_j)$ | Fair: Doesn't like if the partner performs worse or better |

Table 2: Demonstration of reflected personalities by varying the parameters $a$ and $b$ from Equation 1. The variants used in this work are highlighted in blue.

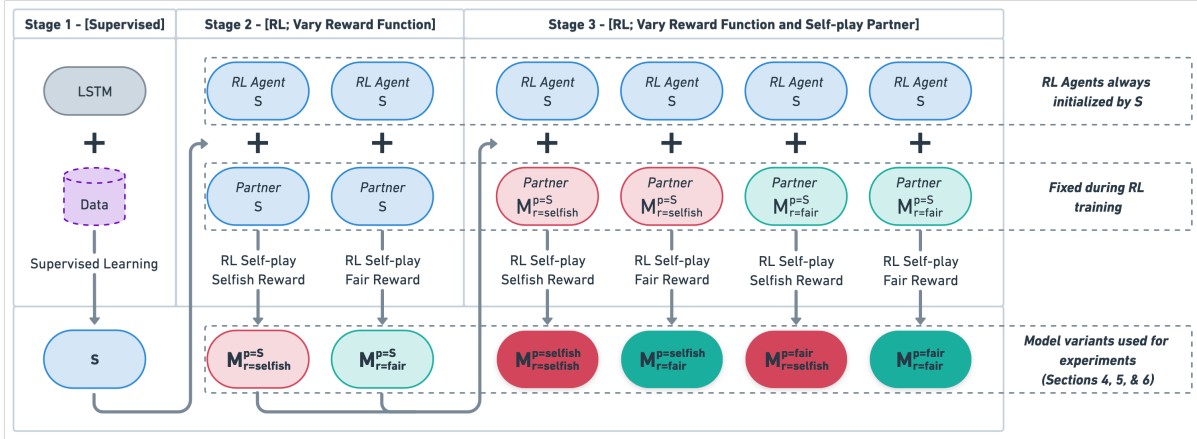

Figure 1: The three-stage process used to design the 6 dialogue agents for our 2 x 3 study. $r$: Reward that the RL agent is trained to maximize. $p$: The partner with which the RL agent is trained. $p=S$ corresponds to the model trained in Stage 1, while $p$=selfish and $p$=fair correspond to the respective models trained in Stage 2.

notations in Figure 1[3].

**Hyperparameters**: We borrowed the hyperparameters from Lewis et al. (2017) and refer the readers to that paper for full details. The supervised model is trained for 30 epochs with a batch size of 16 using stochastic gradient descent. The initial learning rate is kept as 1.0, clipping gradients with $L^2$ norm exceeding 0.5. This was followed by annealing of the learning rate by a factor of 5 per epoch. All the dialogue agents used in the experiments are initialized from this supervised model and trained for nearly 16k agent-agent interactions with the partner model, using a learning rate of 0.1 and a discount factor of $\gamma$=0.95. We use a length cut-off of 20 utterances to simulate walkaways: if a dialogue reaches 20 utterances, this is seen as a disagreement, and both players end up with 0 points.

**Human Evaluation**: We performed a human evaluation on the Prolific[4] crowdsourcing platform. We collected nearly 100 agent-human conversations for each of the 6 dialogue models, where one hu-

man worker was allowed to participate only once. The workers were paid a base payment for their time, along with a lottery-based bonus that was dependent on their performance and effort. We provide more details in Appendix B, including statistics, worker qualifications, payments, and the design of the user interface.

## 5 Results

Table 3 summarizes the human evaluation results. We analyze 3 key metrics: the points scored by the human, by the agent, and the total joint points – an indicator of the total value created in the negotiation. We also report the %age of walkways (%age of dialogues that do not reach an agreement). We discuss the significant trends below.

To analyze the overall performance, we conducted 2 (reward $r$: selfish vs. fair) x 3 (partner $p$: supervised vs. selfish vs. fair) ANOVAs on the points earned in the negotiation. First, we found no significant differences in the points earned by the dialogue agents. However, the agent reward $r$ significantly affected human points (F(1, 577) =

---

[3]Our implementation is based on https://github.com/facebookresearch/end-to-end-negotiator.

[4]https://www.prolific.co/

| Model | Points Scored (Including walkways) ↑ | | | Points Scored (Excluding walkways) ↑ | | | Walkaways ↓ |
|---|---|---|---|---|---|---|---|
| | Human | Agent | Joint | Human | Agent | Joint | (in %) |
| $M_{r=fair}^{p=S}$ | 5.72 (0.29) | 5.99 (0.29) | 11.71 (0.43) | 6.03 (0.28) | 6.32 (0.26) | 12.35 (0.34) | **5.15** |
| $M_{r=fair}^{p=fair}$ | 5.87 (0.29) | **6.04 (0.28)** | 11.91 (0.43) | 6.24 (0.26) | 6.43 (0.25) | 12.67 (0.33) | 6.00 |
| $M_{r=fair}^{p=selfish}$ | 5.59 (0.31) | 5.80 (0.32) | 11.39 (0.42) | 5.89 (0.30) | *6.12 (0.30)* | *12.01 (0.34)* | **5.15** |
| $M_{r=selfish}^{p=S}$ | 4.70 (0.32) | 5.58 (0.39) | 10.28 (0.61) | *5.86 (0.27)* | **6.96 (0.33)** | 12.82 (0.38) | 19.79 |
| $M_{r=selfish}^{p=fair}$ | *4.59 (0.35)* | *5.20 (0.42)* | *9.79 (0.67)* | 6.07 (0.29) | 6.88 (0.37) | 12.96 (0.41) | *24.44* |
| $M_{r=selfish}^{p=selfish}$ | **6.18 (0.30)** | 5.90 (0.28) | **12.09 (0.48)** | **6.85 (0.25)** | 6.54 (0.23) | **13.39 (0.31)** | 9.71 |

Table 3: Results from the human evaluation study. We report the Mean (Standard Error) wherever applicable. The **Joint** points are scored by computing the mean over the sum of the points scored by both players – an indicator of the joint value created in the negotiation. The maximum possible points for a player in a negotiation is 10. ↑: Higher is better, ↓: Lower is better. In each column, we highlight the worst and the best scores in **red** and **blue** respectively. We discuss the significant trends in Sections 5 and 6.

5.00, p = .03), such that human partners playing with fair agents ($r$=fair) earned more points (M = 5.73; SE = 0.18) than those playing with selfish ones (M = 5.16; SE = 0.18). There was also a main effect of the partner $p$ (F(2, 577) = 3.09, p = .046), but both of these main effects were qualified by a significant interaction (F(2, 577) = 5.40, p = .005). Consequently, this led to similar significant trends in the joint points earned (F(1, 577) = 5.21, p = .02), such that fair agents ($r$=fair) earned more joint points with their partner (M = 11.67; SE = 0.29) than selfish ones (M = 10.72; SE = 0.29).

Interestingly, human partners earned more points with $M_{r=selfish}^{p=selfish}$ agent compared to other selfish agents, which also led to more joint points, bringing it on par with (or even better than) fair agents. A plausible explanation is that since the $M_{r=selfish}^{p=selfish}$ agent is trained with an uncompromising partner (unlike other agents with $r$=selfish), it is better exposed to the repercussions of not making concessions for the partner since the agent receives a 0 reward if there is no agreement (within 20 utterances). Thus, the agent learns to "*give in*" in order to avoid no agreements. Next, we test this explicitly by analyzing the %age of walkaways for each agent.

**%age of walkaways**: Indeed, a log-linear analysis reveals a significant interaction between the agent's reward $r$ and partner $p$ on the %age of walkaways ($G^2$ = 31.1, p < .0001). Specifically, the effect of partner $p$ only appears among the selfish agents ($r$=selfish), as revealed by a Chi-squared test ($\chi^2(2)$= 7.63, p = .02), where only 9.7% of conver-

sations with $M_{r=selfish}^{p=selfish}$ agent end without an agreement, much lower than with other selfish agents. This effect was not significant among the agents trained with a fair reward, in which case, the %age of walkaways is observed to be low, regardless of the partner $p$.

| Context | |
|---|---|
| Counts | Book = 1, Hat = 3, Ball = 1 |
| Model Values | Book = 2, Hat = 1, Ball = 5 |
| Human Values | Book = 10, Hat = 0, Ball = 0 |
| **Dialogue** | |
| Human | I would like the books |
| Model | can i have the ball and the book and you can have the hats |
| Human | you can have the ball and let's split the books |
| Model | how about i get the ball and two hats and you get the rest ? |
| Human | i am ok with that |
| Model | <dealselection> |
| **Output** | |
| Model | Book = 0, Hat = 2, Ball = 1 |
| Human | Book = 1, Hat = 1, Ball = 0 |
| **Reward** | |
| Model | 7/10 |
| Human | 10/10 |

Table 4: Example conversation between the $M_{r=selfish}^{p=selfish}$ agent and a human partner in our experimental study. The agents helps to find a solution that leads to high performance for both players.

**Removing walkaways**: Once the instances that end up in walkaways are removed, we find that selfish agents ($r$=selfish) earn more points for themselves (M = 6.79; SE = 0.17) than fair agents (M = 6.28; SE = 0.16; F(1, 510) = 4.62, p = .03). This means that the lack of significant effects above in agent points was due to walkaway instances that

result in 0 points for the agent. Further, we note that even when walkaways are removed, the human partners earn more points with $M_{r=\text{selfish}}^{p=\text{selfish}}$ agent than with other selfish agents. We observed similar trends for joint points as well, with maximum joint points for the $M_{r=\text{selfish}}^{p=\text{selfish}}$ agent. This suggests that besides contributing to lesser walkaways, $M_{r=\text{selfish}}^{p=\text{selfish}}$ agent further learns to discover creative solutions that help both the players. We show one such example in Table 4 and provide more examples from the human evaluation in Appendix C.

# 6 Discussion

Going beyond the typical reward formulations used in the literature, this is the first instance of leveraging prior Economics theories to explicitly incorporate the partner performance within the reward of the selfplay RL negotiation agent. Our formulation provides a systematic and general way to train mixed-motive agents with diverse personalities (Table 2). As shown in Figure 1, our multi-stage training process provides an automated way to simulate diverse partner behaviors as well, instead of the unscalable rule-based approaches followed in prior work (for instance, the price-based rules defined for buyer-seller negotiations in Yang et al. (2021)).

The overall points scored in Table 3 show that all fair agents ($r$=fair) and the $M_{r=\text{selfish}}^{p=\text{selfish}}$ agent perform superior to the $M_{r=\text{selfish}}^{p=\text{S}}$ agent, which is trained following the standard procedure used in prior work – in terms of the human points, agent points, and (consequently) the joint points. This suggests that both strategies of varying the reward and varying the partner during RL training show promise for teaching the mixed-motive nature of negotiations to the dialogue agents.

We especially note the superior performance of $M_{r=\text{selfish}}^{p=\text{selfish}}$ agent. Trained with a simplistic reward that maximizes its own performance, $M_{r=\text{selfish}}^{p=\text{selfish}}$ learns to make concessions implicitly by being better exposed to the repercussions of not doing so during training. This observation aligns with the philosophy of the '*Invisible Hand*' in Economics by *Adam Smith* (Grampp, 2000), which suggests that self-interested players are implicitly led (as if by an invisible hand) to cooperate and take actions that benefit others.

## 6.1 Automated Evaluation

To gain additional insights into the behavioral diversity and the performance of the dialogue agents,

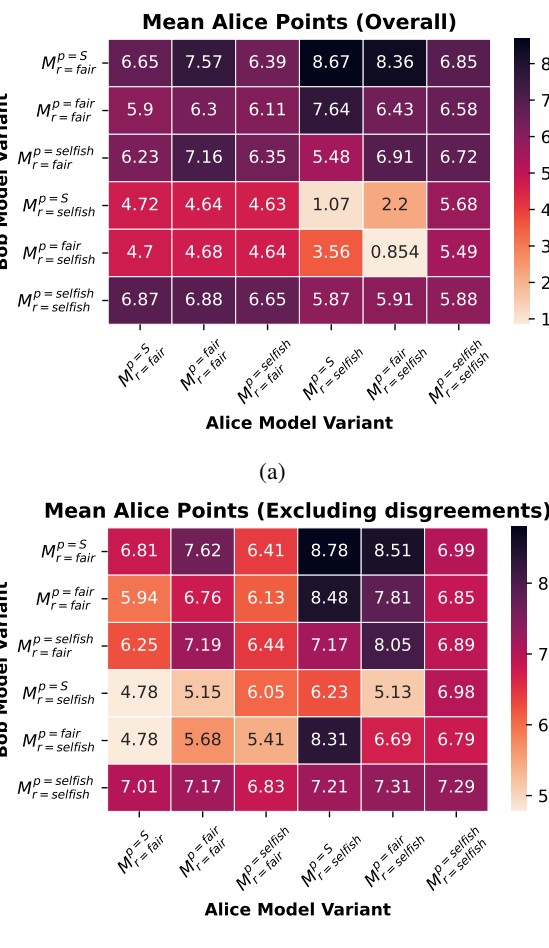

Figure 2: Heatmaps depicting the results from 388 agent-agent interactions. Each cell denotes the points scored (out of 10) by the Alice variant (X-Axis) when it interacts with the corresponding Bob model (Y-Axis).

we analyze the results from the agent-agent interactions. For this purpose, we gather 388 conversations for every pair of agents and observe the average points scored by both agents separately and jointly. We depict the agent performance using heatmaps in Figure 2. Self-interested agents that are less exposed to walkaways during training ($M_{r=\text{selfish}}^{p=\text{S}}$ and $M_{r=\text{selfish}}^{p=\text{fair}}$) tend to exploit the agents trained with a fair reward. However, this behavior backfires when the partner model behaves similarly in a self-interested manner – both agents show uncompromising behavior that leads to higher disagreements (stuck in negotiation for >= 20 utterances) and ultimately, extremely low overall scores.

In general, we find the $M_{r=\text{selfish}}^{p=\text{selfish}}$ agent to be superior, consistently achieving a high performance for itself (the last column) while also enabling a high performance for its partners (the last row). This trend is also evident from the corresponding

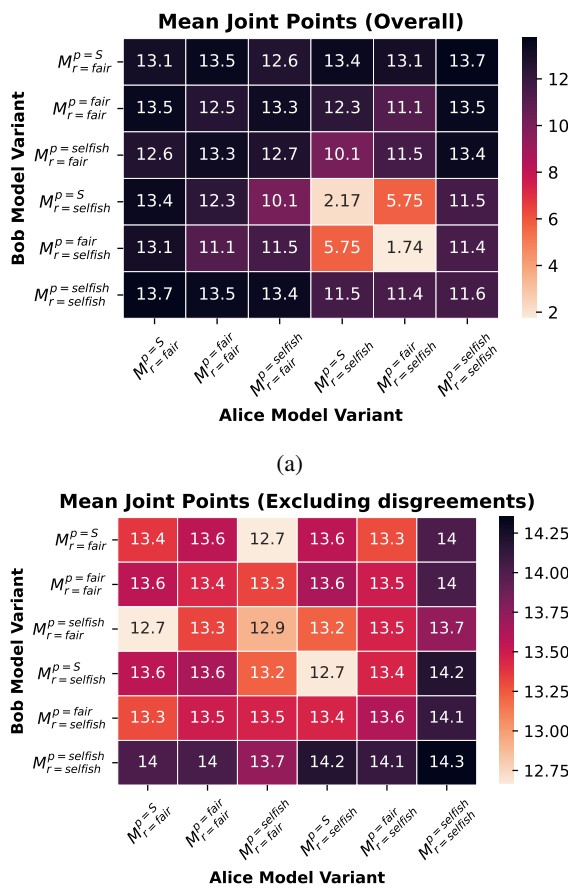

(a)

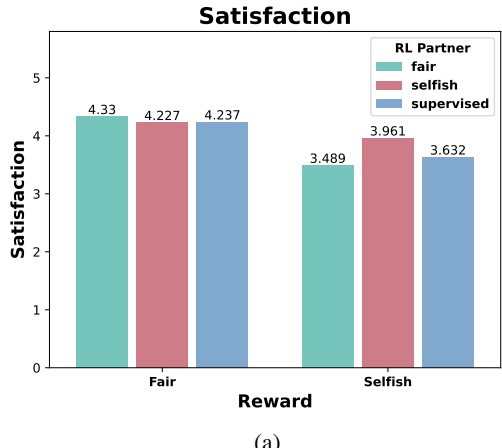

(a)

Figure 4: Subjective assessment by humans. Both metrics are measured on a scale of $1$ to $5$.

(b)

Figure 3: Heatmaps depicting the results from $388$ agent-agent interaction. Each cell denotes the mean joint points scored by the corresponding Alice model variant (X-Axis) and the Bob variant (Y-Axis).

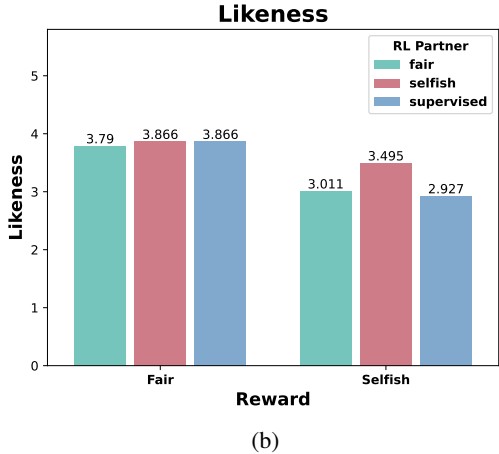

(b)

heatmaps for joint points shown in Figure 3.

## 6.2 Subjective Assessment

Prior work has argued the importance of incorporating subjective measures in social influence tasks like negotiations (Aydoğan et al., 2020). Although this is more relevant for repeated interactions between the same players (unlike in our case, which only involves one negotiation between an agent and a human partner), nevertheless, we present results on the subjective assessment of the human partners for completeness. Through a post-survey, we measured the human partners' satisfaction with the outcome and likeness towards the agent on a five-point scale (more details in Appendix B). We summarize the results in Figure 4.

Based on 2 x 3 ANOVAs, we find that human partners of the fair agents ($r$=fair) were significantly more satisfied (F$(1, 576) = 47.32$, p $< .0001$) as compared to the humans who interacted with the selfish ones, but this was qualified by a marginally significant interaction with the partner $p$ (F$(2, 576)$ $= 2.54$, p $= .08$). This can be attributed to the previously noted observation that human partners, on average, secured more points with fair agents.

We find similar trends with likeness towards the agent as well – human partners report higher likeness when playing with fair agents as compared to selfish ones (F$(1, 577) = 53.95$, p $< .0001$). Interestingly, among the selfish agents ($r$=selfish), the $M_{r=\text{selfish}}^{p=\text{selfish}}$ achieved the highest subjective assessment from the human partners, bringing it close to the performance of fair agents, even though it was trained with a selfish reward.

## 6.3 Measuring Success

As discussed in prior work (Chawla et al., 2023), our analysis reflects upon the multi-faceted nature of the notion of success in negotiations, where observing a single dimension can be misleading. For example, when interacting with model $S$, the

$M_{r=\text{selfish}}^{p=S}$ agent seems to get high points for itself. However, our analysis shows that this is simply due to fewer walkaways, which occur far more often with other selfish agents or human partners. Thus, we stress the importance of a comprehensive evaluation of negotiation dialogue systems.

Perhaps the downstream application context can guide what metrics should be prioritized. From a pedagogical perspective, training agents that accurately reflect the diversity in human behavior (as in this work based on Equation 1) can itself be highly valuable for social skills training. Similarly, subjective assessment of the dialogue agents can be more important in scenarios involving relationships for long-term or repeated social influence interactions.

If the goal is to design a dialogue agent that performs the best for itself (regardless of partner performance), such as in a game context, perhaps the best strategy is to train it with a variety of partner personalities. The agent must develop a *theory-of-mind* about the partner and learn to weigh *extracting concessions* vs. *making concessions* based on the personality of the specific partner in the interaction. We attempted to train such an agent, but unfortunately, not keeping the partner model fixed makes the training process unstable (also observed in Lewis et al. (2017)). One explanation for this is the relatively short conversations in DealOrNoDeal, which makes it hard to infer the partner's personality implicitly. Hence, there is value in extending our analysis to other negotiation dialogue datasets (Yamaguchi et al., 2021; Chawla et al., 2021). In the future, we plan to integrate RL-based planning with Large Language Models (LLMs) for tackling these more complex scenarios, consisting of longer conversations and richer contexts.

# 7 Conclusion

We devised two variations of the standard self-play RL technique to inculcate the mixed-motive nature of negotiation into the dialogue agents. The first approach worked by varying the reward function and thereby, by explicitly pushing the model to take the partner's performance into account. In the second approach, we modified the personality of the partner agent during training, which allowed the RL agent to discover the mixed-motive nature of the task implicitly.

We find that both techniques hold promise, with an especially strong performance from the agent that is trained with a selfish reward and a self-

interested partner. This agent not only improves on the agreement rate but also learns to discover offers that create value for its partner without hurting its own points significantly.

# 8 Broader Impact and Ethical Considerations

## 8.1 Dataset Used

We used a publicly available version of the DealOrNoDeal dataset[5]. The dataset was completely anonymized prior to its release by the authors. Moreover, we verified the licensing details to ensure that the dataset was used only within its intended scope.

## 8.2 Human Evaluation

Our human evaluation experiment was approved by the relevant Institutional Review Board (IRB). Before the data collection, each participant signed an Informed Consent document, which outlined the study's objectives, warned about potential discomfort, and acknowledged the collection and future use of data. The participants were also informed of their right to withdraw from the study at any time. Furthermore, they were instructed to refrain from using offensive or discriminatory language during the experiment. The compensation provided to participants adhered to the guidelines established by our IRB approval process. Lastly, any mention of the personality of the human participants in this paper is based on the standard procedures of collecting personality metrics in the literature.

## 8.3 Automatic Negotiation Systems

Negotiation has been actively studied in diverse research areas, including Economics, Psychology, and Affective Computing (Carnevale and Pruitt, 2003). More recently, it has been studied as a social influence dialogue task for automated systems (Chawla et al., 2023).

Automated systems capable of negotiating via realistic modes of communication, such as natural language, hold a huge potential in making social skills training more scalable and effective (Johnson et al., 2017). Personality-based variants of dialogue systems (such as the ones explored in this work) can also help to design experimental studies in Psychology to better understand human decision-making (Gratch et al., 2015). Further, the

---

[5]https://github.com/facebookresearch/end-to-end-negotiator

techniques developed can help to advance conversational AI, such as the Google Duplex (Leviathan and Matias, 2018), a system that engages in a simple form of negotiation to book a haircut appointment over the phone.

While these use cases are encouraging, these systems must be deployed in the wild by following proper ethical guidelines. Our primary recommendation is maintaining transparency – not only about the identity of the system but also about its capabilities, key design objectives, the data on which the model has been fine-tuned, along with any known discriminative or other undesirable behaviors. We encourage rigorous testing of the model behaviors pre-deployment and continuous monitoring post-deployment. We believe these recommendations should be followed for any human-centric AI models, including social influence dialogue systems and even Large Language Models.

## 9 Limitations

**Task Design**: The DealOrNoDeal task is based on a simplified abstraction of real-world negotiations, referred to as the Multi-Issue Bargaining Task or MIBT (Fershtman, 1990). MIBT assumes a fixed set of issues and predefined priorities for players before the negotiation begins. Although popular in NLP research and beyond, the MIBT framework does not capture several realistic negotiation scenarios, such as complex cases where an item can be split into more than one unit or cases where the priorities of the negotiators change during the interaction. Future work in data collection for negotiation tasks should consider such scenarios. Among the available datasets that use MIBT, more recent datasets capture richer negotiation contexts with relatively longer interactions, such as campsite negotiations in the CaSiNo dataset (Chawla et al., 2021) and salary negotiations in the JobInterview dataset (Yamaguchi et al., 2021) - We encourage future work to explore incorporating agent personalities for these datasets.

**Human Evaluation**: Following the design of the DealOrNoDeal dataset that contains dialogues in English, our human evaluation involved workers from a restricted demographic pool – nationality as USA and English as the native language. However, prior research has noted differences in negotiation behaviors across cultures (Andersen et al., 2018; Peng, 2008). Hence, it is unclear if our findings from the human evaluation would directly apply

to workers from a different demographic. While this is out of the scope of our paper, this should be better explored in the future.

## Acknowledgments

We want to thank our colleagues at the University of Southern California for all the comments and helpful discussions that have shaped this project. We also thank the anonymous EMNLP 2023 reviewers for their valuable time and feedback. Our research was sponsored by the Army Research Office and was accomplished under Cooperative Agreement Number W911NF-20-2-0053. The views and conclusions contained in this document are those of the authors and should not be interpreted as representing the official policies, either expressed or implied, of the Army Research Office or the U.S. Government. The U.S. Government is authorized to reproduce and distribute reprints for Government purposes, notwithstanding any copyright notation herein.

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

## A  Problematic self-interested behavior

Figure 5 shows a sample dialogue between two self-interested RL agents based on the standard self-play RL training procedure followed in prior work (Lewis et al., 2017; He et al., 2018). To the best of our knowledge, such an analysis has not been shown in the past. The dialogue provides evidence that the agents fail to learn about the value of compromise in a negotiation. Since both agents are *unwilling to make concessions for their partner*, they simply get stuck.

## B  Human Evaluation Setup

**Crowdsourcing platform**: We performed our initial experiments on Amazon Mechanical Turk (AMT)[6]. However, we faced severe quality issues even with strict worker qualifications and extensive checks in place. We eventually collected our entire data on the Prolific crowdsourcing platform[7], where we could receive a relatively much better data quality.

**Study Design**: Our study involved a pre-survey based on Qualtrics[8] which included attention checks and a personality test to measure the Social Value Orientation (Van Lange et al., 1997)

[6]https://www.mturk.com/
[7]https://www.prolific.co/
[8]https://www.qualtrics.com/

| Context (Alice & Bob: Standard RL Agents) | |
|---|---|
| Counts | Book = 1, Hat = 2, Ball = 2 |
| Alice Values | Book = 2, Hat = 3, Ball = 1 |
| Bob Values | Book = 2, Hat = 1, Ball = 3 |
| **Dialogue** | |
| Alice | i would like the hats and the balls . |
| Bob | i can take the balls but i need the book or both hats |
| Alice | i need the book and at least one other item |
| Bob | i can not make that deal . i need the book and at least 1 hat or a ball |
| ... | ... |
| Bob | i can't do that if i get the book , you can have the rest |
| | *Turn limit reached* |
| **Output** | |
| Alice | <no_agreement> |
| Bob | <no_agreement> |
| **Reward** | |
| Alice | 0/10 |
| Bob | 0/10 |

Figure 5: A sample negotiation dialogue between two copies of the standard RL agent based on Lewis et al. (2017). The task here is to divide the available books, hats, and balls between the two players. In this case, the agents get stuck – both continuously asking what they want without looking for a compromise.

of the human participants (Prosocial vs Proself)[9]. However, in our study, we observed no significant differences among the agents' performances when interacting with Prosocial or Proself human partners. We also included a mini-tutorial to prepare the participants for their upcoming negotiation with a randomly-chosen agent.

The main negotiation task was set up using the LIONESS framework[10], which was hosted on AWS[11] using a Bitnami LAMP stack[12]. We provide a screenshot from the task in Figure 6.

After the negotiation, we used a post-survey to gather the participants' subjective perceptions. For satisfaction, we asked "*How satisfied are you with the negotiation outcome?*", and for likeness, we asked "*How much do you like your opponent?*". We used a 5-point Likert scale for both questions, from *Extremely dissatisfied (dislike)* to *Extremely satisfied (like)*. For the statistical analysis presented in Section 6, we codified this scale from 1.0 to 5.0, considering both of these metrics as continuous measures.

[9]https://static1.squarespace.com/static/523f28fce4b0f99c83f055f2/t/56c794cdf8baf3ae17cf188c/1455920333224/Triple+Dominance+Measure+of+SVO.pdf
[10]https://lioness-lab.org/
[11]https://aws.amazon.com/
[12]https://bitnami.com/stack/lamp/cloud

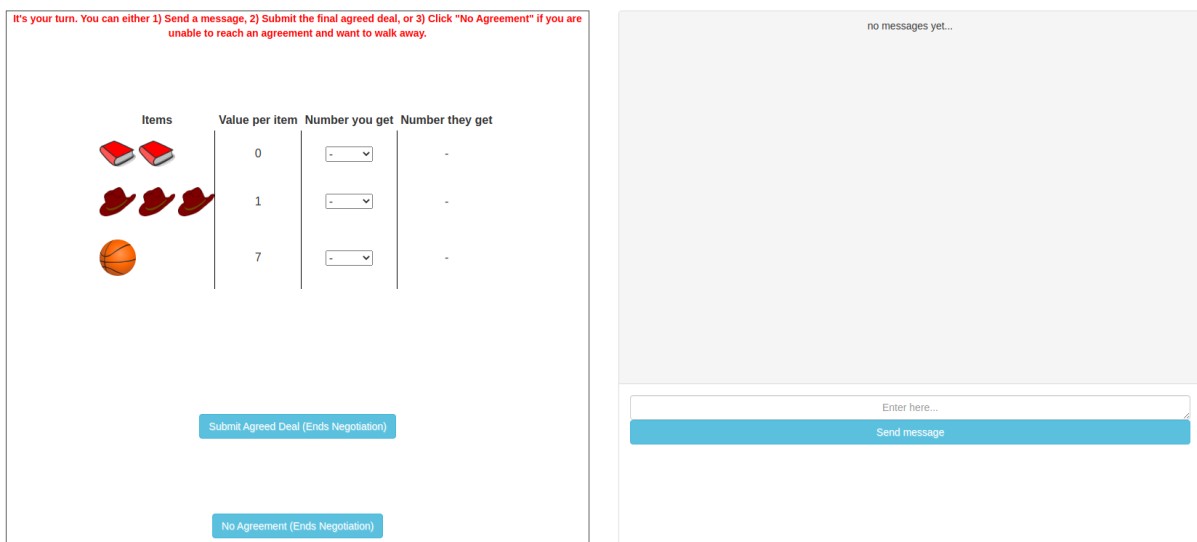

Figure 6: Screenshot from the human evaluation study. The participants first negotiate with a randomly assigned dialogue agent using the chat feature on the right side. Once an agreement is reached, the participant is asked to enter the agreed deal using the options on the left. The participant was also allowed to walk away from the conversation without agreement. The participant was allowed to submit a deal or walk away after at least one turn.

**Worker Qualifications**: Each worker was only allowed to participate once in the study. The worker pool was restricted to the USA, with English as the native language, a minimum approval rate of 90%, and at least 500 minimum number of submissions.
**Worker Payments**: The workers were paid at the rate of $12 per hour. The expected time to complete the study was 10 minutes, resulting in a base pay of 2. In addition, the workers were entered into a lottery where we awarded $10 to 15 randomly selected workers. A worker's chances of winning the lottery depended on their performance and effort put into the task.
**Post-Processing**: For nearly 30 % of the cases, the final deal entered by the human or the agent did not match. However, this disagreement did not mean a disagreement in the negotiation. Instead, this was primarily due to either an error by the model, the human worker, or both (occurs rarely). Hence, we post-processed the data to fix these instances. This was done manually by the authors of the paper (that is, experts knowledgeable about the task). For each instance, either the agreed deal was identified or the instance was discarded from evaluation if the agreed deal was completely unclear (occurs rarely).
**Statistics**: We summarized the statistics in Table 5. We collected nearly 100 agent-human conversations for every dialogue model. In general, we find the conversations to be longer between humans and selfish agents (and more number of words per utterance from the selfish agents), as compared to

the fair ones. This is probably due to the selfish agents negotiating harder for high-value deals for themselves.

## C  Sample conversations from Human Evaluation

We show sample conversations from the human evaluation for all dialogue agents in Tables 6 to 11.

| Model | # of dialogues | Avg. # of utterances/dialogue | Avg. # of words/utterance |
|---|---|---|---|
| $M^{p=S}_{r=fair}$ | 97 | 4.35 | 5.47 |
| $M^{p=fair}_{r=fair}$ | 100 | 4.71 | 5.50 |
| $M^{p=selfish}_{r=fair}$ | 97 | 4.91 | 5.31 |
| $M^{p=S}_{r=selfish}$ | 96 | 8.74 | 6.94 |
| $M^{p=fair}_{r=selfish}$ | 90 | 6.50 | 6.75 |
| $M^{p=selfish}_{r=selfish}$ | 103 | 6.29 | 6.00 |

Table 5: Statistics from the human evaluation study.

| Context | |
|---|---|
| Counts | Book = 3, Hat = 1, Ball = 2 |
| Model Values | Book = 2, Hat = 2, Ball = 1 |
| Human Values | Book = 0, Hat = 8, Ball = 1 |
| **Dialogue** | |
| Human | Hi there. How about you get both the balls, I get 2 books and 1 hat? |
| Model | deal |
| Human | <dealselection> |
| **Output** | |
| Model | Book = 1, Hat = 0, Ball = 2 |
| Human | Book = 2, Hat = 1, Ball = 0 |
| **Reward** | |
| Model | 4/10 |
| Human | 8/10 |

Table 6: Sample human evaluation output for $M^{p=fair}_{r=fair}$

| Context | |
|---|---|
| Counts | Book = 2, Hat = 3, Ball = 1 |
| Model Values | Book = 0, Hat = 1, Ball = 7 |
| Human Values | Book = 2, Hat = 2, Ball = 0 |
| **Dialogue** | |
| Model | i would love the ball and the books |
| Human | Okay I want all the hats then. |
| Model | <dealselection> |
| **Output** | |
| Model | Book = 2, Hat = 0, Ball = 1 |
| Human | Book = 0, Hat = 3, Ball = 0 |
| **Reward** | |
| Model | 7/10 |
| Human | 6/10 |

Table 8: Sample human evaluation output for $M^{p=S}_{r=fair}$

| Context | |
|---|---|
| Counts | Book = 1, Hat = 4, Ball = 2 |
| Model Values | Book = 4, Hat = 0, Ball = 3 |
| Human Values | Book = 0, Hat = 2, Ball = 1 |
| **Dialogue** | |
| Model | i need the book and 2 balls |
| Human | Not possible, I need 1 ball and 2 hats |
| Model | ok |
| Human | Ok, deal. |
| Model | <dealselection> |
| **Output** | |
| Model | Book = 1, Hat = 2, Ball = 1 |
| Human | Book = 0, Hat = 2, Ball = 1 |
| **Reward** | |
| Model | 7/10 |
| Human | 5/10 |

Table 7: Sample human evaluation output for $M^{p=selfish}_{r=fair}$

| Context | |
|---|---|
| Counts | Book = 2, Hat = 2, Ball = 3 |
| Model Values | Book = 3, Hat = 2, Ball = 0 |
| Human Values | Book = 2, Hat = 0, Ball = 2 |
| **Dialogue** | |
| Model | i would like the hats and books . |
| Human | I want the two books, you take the rest. |
| Model | i will take the balls and books |
| Human | sorry, that's no deal. |
| Model | <dealselection> |
| Human | <walkaway> |
| **Output** | |
| Model | <walkaway> |
| Human | <walkaway> |
| **Reward** | |
| Model | 0/10 |
| Human | 0/10 |

Table 9: Sample human evaluation output for $M^{p=S}_{r=selfish}$

| Context | |
|---|---|
| Counts | Book = 1, Hat = 1, Ball = 4 |
| Model Values | Book = 8, Hat = 2, Ball = 0 |
| Human Values | Book = 5, Hat = 1, Ball = 1 |
| **Dialogue** | |
| Model | i would like the book and the hat . |
| Human | Hey! You would like one book and the hat? |
| Model | i would like the book and the hat . |
| Human | I will give you the book, but I'd like the hat and the balls to make sure we both get 5. |
| Model | i could use the book and the hat . |
| Human | Do you agree? |
| Model | <dealselection> |
| Human | <walkaway> |
| **Output** | |
| Model | <walkaway> |
| Human | <walkaway> |
| **Reward** | |
| Model | 0/10 |
| Human | 0/10 |

Table 10: Sample human evaluation output for $M_{r=\text{selfish}}^{p=\text{fair}}$

| Context | |
|---|---|
| Counts | Book = 1, Hat = 4, Ball = 2 |
| Model Values | Book = 0, Hat = 2, Ball = 1 |
| Human Values | Book = 4, Hat = 0, Ball = 3 |
| **Dialogue** | |
| Human | hi, what items do you need? |
| Model | i want the hats and the balls |
| Human | I can give you all the hats only |
| Model | i need the hats and the book |
| Human | I can only offer the hats |
| Model | i can't do that |
| Human | I can give you all the hats and one ball |
| Model | i can't do that . i need the hats and at least 1 ball |
| Human | yeah i can do that |
| Model | <dealselection> |
| **Output** | |
| Model | Book = 1, Hat = 0, Ball = 1 |
| Human | Book = 0, Hat = 4, Ball = 1 |
| **Reward** | |
| Model | 9/10 |
| Human | 7/10 |

Table 11: Sample human evaluation output for $M_{r=\text{selfish}}^{p=\text{selfish}}$