# OpenReview forum: "Be Selfish, But Wisely: Investigating the Impact of Agent Personality in Mixed-Motive Human-Agent Interactions"
_EMNLP/2023/Conference — EMNLP 2023 Main_

### Official Review · Reviewer_atVW · 2023-07-25

**Soundness:** 3

**Excitement:**

4: Strong: This paper deepens the understanding of some phenomenon or lowers the barriers to an existing research direction.

**Justification For Ethical Concerns:**

No separate review is needed.

**Missing References:**

No missing references.

**Paper Topic And Main Contributions:**

The paper focuses on designing agents for mixed-motive interactions. Two variations of the standard self-play Reinforcement Learning technique are proposed to show the mixed-motive nature of negotiation into the dialogue agents by varying the reward function and the personality of the partner agent during training.

Main contributions:
-It is shown that the standard selfplay RL training procedure fails to develop sophisticated negotiation dialogue systems that
are useful in practical scenarios.
-Novel ways to modify the training procedure are proposed to design systems that exhibit diverse personalities and better understand the concept of walkaways.
-Automated and human evaluations are carried out in order to investigate what model variation allows for superior performance.


**Questions For The Authors:**

No questions.

**Reasons To Accept:**

Aiming to design agents with diverse personalities, the authors modify the training procedure in two novel ways.
The key finding of the paper is that  a selfish agent which maximizes its own performance and avoids walkaways, performs better than other variants by implicitly learning to generate value for both itself and the negotiation partner.
The implications are discussed for designing and evaluating negotiation dialogue systems in the future.
Limitations in task design and human evaluation are shown which can be taken into account in further work.

**Reasons To Reject:**

The paper should be accepted.

**Reproducibility:**

3: Could reproduce the results with some difficulty. The settings of parameters are underspecified or subjectively determined; the training/evaluation data are not widely available.

**Reviewer Confidence:**

3: Pretty sure, but there's a chance I missed something. Although I have a good feel for this area in general, I did not carefully check the paper's details, e.g., the math, experimental design, or novelty.

**Typos Grammar Style And Presentation Improvements:**

No  typographical or grammatical errors have been found.
About organization of the paper: Appendix (pages 11 to 15, starting from line 385) includes important results that actually belong to the main part of the paper (Figures 4 to 6, Tables 6 to 11).

---

> ### Author Rebuttal · Authors · 2023-08-26
>
> Thank you for your feedback. We are glad that we were able to get our key contributions across.
>
> **On improving the paper organization**: The Appendix only contains additional information that is not central to our paper, such as precise details of the human evaluation experiment, example dialogues, and supporting figures. Nevertheless, if the reviewer feels that this information should be in the main paper, we can definitely address this easily. With more space in the camera-ready version, we will include additional details about the human evaluation experiment and Figure 6 within the main paper.

---

### Official Review · Reviewer_qS7U · 2023-07-31

**Typos Grammar Style And Presentation Improvements:** 1. The first sentence in Abstract can…
**Soundness:** 4

**Excitement:**

4: Strong: This paper deepens the understanding of some phenomenon or lowers the barriers to an existing research direction.

**Paper Topic And Main Contributions:**

The paper's main objective is to enhance negotiation dialogues by incorporating agent personality. Existing negotiation systems mainly focus on maximizing their own output, neglecting the potential benefits of making concessions during negotiations. To address this, the paper explores two training strategies: reward function and reinforcement learning partner. The reward function captures the personality's aggressiveness, promoting concession-making or assertiveness. The reinforcement learning partner acts as a teacher model, guiding the proposed model's learning process. By selecting different teacher models, the student model adopts corresponding negotiation strategies. Experimental results demonstrate the effectiveness of both strategies. Automatic and human evaluations reveal that considering the opposite party's earnings and making concessions can help avoid negotiation failure and increase own outputs.

**Reasons To Accept:**

1. It is an interesting topic of investigating how to apply agent personality to a negotiation dialogue and how they could impact on negotiation outputs. The paper includes enough details to reproduce the results for future researcher.
2. The study is solid. The choice of reward function and the experimental results are both backed by economics theories. The finding can be beneficial for researchers to develop more advanced negotiation dialogue systems.

**Reasons To Reject:**

1. The model and the task are rather simple. The negotiation dialogues are also simple in language and short in length. They make the resulting finding relatively straightforward and prevent the model from coming up with more complicated strategies.
2. The experiments are conduct on a single dataset/task, making the finding less convincing. It is not sure if the "selfish-but-wise" negotiation strategy is also preferable on other negotiation tasks.

**Reproducibility:**

4: Could mostly reproduce the results, but there may be some variation because of sample variance or minor variations in their interpretation of the protocol or method.

**Reviewer Confidence:**

3: Pretty sure, but there's a chance I missed something. Although I have a good feel for this area in general, I did not carefully check the paper's details, e.g., the math, experimental design, or novelty.

---

> ### Author Rebuttal · Authors · 2023-08-26
>
> Thank you for your feedback. We are glad that you found our paper to be interesting, thorough, and beneficial for future work.
>
> “The model and the task are rather simple …”, “The experiments are conducted on a single dataset/task …”
>
> 1) Our work improves upon the standard selfplay RL approach for mixed-motive negotiation dialogues. This approach was introduced by Lewis et al. 2017 [1] on the DealOrNoDeal task/dataset (DND), and hence, DND was also used in our experiments.
>
> 2) Besides, DND does capture a variety of negotiation scenarios, depending on how aligned the priorities of the two players are. This has made it a popular choice for research in this area.
>
> 3) We note that following Lewis et al., our RL models operate at utterance-level (one action = generating the next utterance). For tasks with longer conversations, prior work has instead relied on RL models that work at the dialogue act level (one action = predict the next dialogue act). This is still an under-explored direction for complex negotiation dialogues. Our approach can be naturally extended to this setting (using a similar reward design and partner models), but we believe that making this connection is beyond the scope of this paper. In fact, our ongoing work is exploring how to combine our approach with LLMs for more complex negotiation scenarios.
>
> Note that we have already discussed this briefly in the paper - in Section 2 (Lines 180-186), Section 6 (Lines 557 - 566), and Section 9 (Lines 641 - 661).
>
> **Presentation Improvements**: Thank you for pointing these out. We will fix them in the final version.
>
> References
>
> [1] Deal or No Deal? End-to-End Learning of Negotiation Dialogues (Lewis et al., EMNLP 2017)

---

### Official Review · Reviewer_BeTN · 2023-08-05

**Soundness:** 4

**Excitement:**

4: Strong: This paper deepens the understanding of some phenomenon or lowers the barriers to an existing research direction.

**Paper Topic And Main Contributions:**

The authors train multiple agents on the Deal or No Deal negotiation task by varying the RL reward function. The authors take inspiration from negotiation theory to derive a reward function that models a broad range of behaviors like selfishness, envy and fairness. They then train partners and agents using these reward functions to form a grid of agents. Each agent is evaluated using automated metrics and human evaluations. Notably, the study reveals that human evaluators perceive comparable behavior between selfishly trained models and those trained with fair rewards, only when selfish agents play against each other.

**Reasons To Accept:**

A1. Thoroughness: Authors evaluate the agents via automated metrics and human assessment, and perform a thorough analysis of all results.

A2. Interesting insights: Authors find academically intriguing insights into negotiation behaviors with real-world parallels, and make connections to Adam Smith's "The Invisible Hand"

**Reasons To Reject:**

R1. Novelty Clarification: The work predominantly involves a comprehensive evaluation, primarily modifying the reward function and training partner. While I don't consider this a problem, the authors claim novelty in training procedure in line 126.

**Reproducibility:**

4: Could mostly reproduce the results, but there may be some variation because of sample variance or minor variations in their interpretation of the protocol or method.

**Reviewer Confidence:**

3: Pretty sure, but there's a chance I missed something. Although I have a good feel for this area in general, I did not carefully check the paper's details, e.g., the math, experimental design, or novelty.

---

> ### Author Rebuttal · Authors · 2023-08-26
>
> Thank you for your feedback. We are glad that you found our paper to be thorough and interesting.
>
> Clarification on the novelty of the training procedure: We stress two key aspects of novelty here:
>
> 1) **A new reward formulation**: To the best of our knowledge, this is the first instance of leveraging prior Economics theories to explicitly incorporate the partner performance within the reward of the selfplay RL negotiation agent. Our formulation provides a systematic and general way to train mixed-motive agents with diverse personalities (Table 2). Further, this is impactful since this goes beyond the typical reward formulation in the literature (e.g., [1], [2], and [3]).
>
> 2) **An automatic way to vary the partner personality**: Prior work in partner modeling has relied on hand-crafted rules to define diverse opponent populations (e.g., [3]). These rules are easy to define in case of single-issue price-based negotiations but are not scalable to multi-issue scenarios with more complex utterance structures. Our work provides an automated way to simulate diverse partner personalities (Figure 1) that can fuel future work in this direction.
>
> We will elaborate on these points in Sections 1 and 2.
>
> References
>
> [1] Deal or No Deal? End-to-End Learning of Negotiation Dialogues (Lewis et al., EMNLP 2017)
>
> [2] Decoupling Strategy and Generation in Negotiation Dialogues (He et al., EMNLP 2018)
>
> [3] Improving Dialog Systems for Negotiation with Personality Modeling (Yang et al., ACL-IJCNLP 2021)

---

### Meta-Review · Area_Chair_sTh4 · 2023-09-17

**Recommendation:** 5

**Metareview:**

In general, all reviewers agreed that this was sound and exciting work.  Only minor issues were raised in the reviews, as described below.

**Summary of Reviewer Feedback and Discussion:**
- **Reviewer BeTN** liked that the authors thoroughly evaluated multiple agents using both automated metrics and human assessment, and that they also thoroughly analyzed all results.  They also found that the insights resulting from the work were academically interesting.  However, although the authors claimed that their training procedure was novel, Reviewer BeTN was unconvinced of its novelty.  In their rebuttal, the authors clarified that their training procedure was novel in two ways: it introduces a new reward formulation, as well as an automated technique for varying partner personality.  They promised to elaborate on this in the updated manuscript.
- **Reviewer qS7U** felt that the paper covered an interesting topic with sufficient detail for future replication.  They also thought that the study was solid and backed by economic theory.  However, they felt that the model and task were rather simple, with short and straightforward negotiation dialogues, preventing the model from coming up with more complicated strategies.  They also noted that the experiments were only conducted on a single dataset and task, which made the reported findings less convincing.  In their rebuttal, the authors elaborated on the complexity of their model, and mentioned that these were briefly discussed in the original manuscript.  They could extend these discussions in the revised manuscript.
- **Reviewer atVW** appreciated the novelty of the work, and felt that the paper thoroughly discussed the implications for designing and evaluating negotiation dialogue systems in the future.  They also liked that the paper clearly explained the limitations associated with the task design and human evaluation.  They felt that some of the results in the appendix should be moved to the main body of the paper.  In their rebuttal, the authors thanked the reviewer and noted that they will move additional details about the human evaluation as well as Figure 6 to the main body of the paper.

---

### Decision · Program_Chairs · 2023-10-07

**Decision:**

Accept-Main

**Comment:**

In general, all reviewers agreed that this was sound and exciting work.  Only minor issues were raised in the reviews, as described below.

**Summary of Reviewer Feedback and Discussion:**
- **Reviewer BeTN** liked that the authors thoroughly evaluated multiple agents using both automated metrics and human assessment, and that they also thoroughly analyzed all results.  They also found that the insights resulting from the work were academically interesting.  However, although the authors claimed that their training procedure was novel, Reviewer BeTN was unconvinced of its novelty.  In their rebuttal, the authors clarified that their training procedure was novel in two ways: it introduces a new reward formulation, as well as an automated technique for varying partner personality.  They promised to elaborate on this in the updated manuscript.
- **Reviewer qS7U** felt that the paper covered an interesting topic with sufficient detail for future replication.  They also thought that the study was solid and backed by economic theory.  However, they felt that the model and task were rather simple, with short and straightforward negotiation dialogues, preventing the model from coming up with more complicated strategies.  They also noted that the experiments were only conducted on a single dataset and task, which made the reported findings less convincing.  In their rebuttal, the authors elaborated on the complexity of their model, and mentioned that these were briefly discussed in the original manuscript.  They could extend these discussions in the revised manuscript.
- **Reviewer atVW** appreciated the novelty of the work, and felt that the paper thoroughly discussed the implications for designing and evaluating negotiation dialogue systems in the future.  They also liked that the paper clearly explained the limitations associated with the task design and human evaluation.  They felt that some of the results in the appendix should be moved to the main body of the paper.  In their rebuttal, the authors thanked the reviewer and noted that they will move additional details about the human evaluation as well as Figure 6 to the main body of the paper.